# Review of Pharmacologic Considerations in the Use of Azole Antifungals in Lung Transplant Recipients

**DOI:** 10.3390/jof7020076

**Published:** 2021-01-22

**Authors:** Megan E. Klatt, Gregory A. Eschenauer

**Affiliations:** Department of Clinical Pharmacy, University of Michigan College of Pharmacy, 428 Church Street, Ann Arbor, MI 48109, USA; klattm@med.umich.edu

**Keywords:** antifungal drugs, azoles, lung transplant, interactions, adverse reactions, therapeutic drug monitoring

## Abstract

Mold-active azole antifungals are commonly prescribed for the prevention of invasive fungal infections in lung transplant recipients. Each agent exhibits a unique pharmacologic profile, an understanding of which is crucial for therapy selection and optimization. This article reviews pharmacologic considerations for three frequently-used azole antifungals in lung transplant recipients: voriconazole, posaconazole, and isavuconazole. Focus is drawn to analysis of drug-interactions, adverse drug reactions, pharmacokinetic considerations, and the role of therapeutic drug monitoring with special emphasis on data from the post-lung transplant population.

## 1. Introduction

Lung transplant recipients are at substantial risk for developing invasive fungal infections (IFIs), with a reported cumulative incidence of 8.6% in the first year after transplantation. In contrast to other solid organ transplant types, in which infections due to *Candida* spp. predominate, *Aspergillus* spp. (44%), in particular *Aspergillus fumigatus*, and other molds (19.8%) cause the majority of IFIs in lung transplant recipients [1]. *Candida* spp., mainly *Candida albicans*, while less frequently observed, also cause a substantial proportion of IFIs. Molds generally cause invasive pulmonary disease, while *Candida* can cause anastomotic tracheobronchitis as well as extrapulmonary infections such as early post-transplant pleural space infection [2,3]. As a result, guidelines from the International Society for Heart and Lung Transplantation (ISHLT), Infectious Diseases Society of America (IDSA), and American Society of Transplantation Infectious Diseases Community of Practice (AST) all recommend post-transplant prophylaxis with a spectrum inclusive of *Aspergillus* spp. Consensus on strategy (targeted vs. universal) and preferred agent (inhaled amphotericin B preparations, systemic mold-active azole agents, or a combination thereof) has not been reached. Guidelines recommend an extended duration of prophylaxis (3–6 months) when it is employed [4,5,6], and in patients who do develop invasive mold infection (IMI), therapy duration can range from 6 weeks to lifelong [5].

The focus of this review is on pharmacologic considerations with the use of mold-active azole antifungals voriconazole, posaconazole, and isavuconazole in lung transplant recipients, given their frequency of use in this population and often extended durations. Thus far, development of azole resistance in *Aspergillus* in lung transplant recipients is uncommon. Given often prolonged durations of azole prophylaxis/treatment, however, resistance may be inevitable and eventually mandate other approaches [7]. Itraconazole is not included as it is not recommended for treatment of aspergillosis and its use is limited by toxicity and pharmacokinetic concerns [5]. The echinocandin antifungals (anidulafungin, caspofungin, and micafungin) are not discussed given their intravenous route of administration and sparse data in lung transplant recipients. Finally, the reader is referred elsewhere for information pertaining to inhaled amphotericin B [8]. Areas addressed include drug–drug interactions, short- and long- term toxicities, and the role of therapeutic drug monitoring.

## 2. Drug–Drug Interactions

All three azole agents are inhibitors of the CYP3A4 enzyme system, and thus are implicated in many drug–drug interactions. Voriconazole additionally inhibits CYP2C8/9 and CYP2C19. Table 1 presents the drug interaction potential and magnitude of effect of these agents.

Understandably, most research has focused on the impact of azole antifungals on the plasma concentrations of cyclosporine, tacrolimus, and sirolimus, given that these agents are metabolized by CYP3A4. The reader is referred to comprehensive reviews regarding interactions in transplant populations for detailed recommendations for management of drug–drug interactions with these agents [9,13]. However, general characteristics can be deduced. First, the magnitude of azole-mediated metabolic inhibition differs among immunosuppressants. In general, cyclosporine is impacted less than tacrolimus and (especially) sirolimus. For example, it is recommended to decrease cyclosporine, tacrolimus, and sirolimus doses by 50%, 66%, and >90%, respectively, when voriconazole is initiated. In fact, voriconazole and sirolimus co-administration is contraindicated [13]. Second, while the above serves as general guidance regarding empirical immunosuppressive dose reduction, there is significant interindividual variability in the magnitude of interaction, precluding a confident, universal approach to management of drug–drug interactions with these agents [14]. Rather, empirical adjustments should be made, as appropriate, for immunosuppressants followed by close therapeutic drug monitoring. Third, while all three azoles inhibit CYP3A4, isavuconazole appears to be a more modest inhibitor compared to the strong inhibitory effect of voriconazole and posaconazole [13]. In fact, investigators at the University of Pittsburgh found a very mild impact on tacrolimus dosing and concentrations when isavuconazole prophylaxis was discontinued in 55 solid organ transplant recipients [10]. As such, while plasma concentrations should be monitored, empirical dose reductions are not recommended with isavuconazole initiation [13].

As isavuconazole and voriconazole are substrates of important CYP3A4 enzyme systems, and posaconazole is not, it may be expected that isavuconazole and voriconazole pharmacokinetics would be impacted by concomitant administration with other inhibitors or inducers of the CYP enzyme system. However, with co-administration of competitive inhibitors, it appears that azole agents more often inhibit the metabolism of other therapies, rather than the reverse [15]. However, significant metabolic inducers can impact all agents, including posaconazole. For example, rifampin decreases voriconazole and isavuconazole area under the curve (AUC) by >90%, but also reduced posaconazole serum concentrations (likely due to induction of metabolism by uridine diphosphate glucuronosyltransferase enzymes) by >50% in a case report [16,17,18].

In conclusion, isavuconazole, posaconazole, and voriconazole are prone to many drug–drug interactions. As such, a robust assessment of potential interactions should be undertaken any time these agents are initiated or discontinued in lung transplant recipients. Tertiary drug reference resources, such as Lexicomp^®^ or Micromedex^®^, provide drug interaction analysis tools that can facilitate such assessment. Transplant and infectious diseases pharmacists can also be consulted to assess the impact of such interactions, as well as therapeutic steps to take to resolve significant concerns. For example, while rifampin co-administration is untenable with voriconazole, rifabutin is a significantly less potent enzyme inducer and a case report describes an approach to enable co-administration with voriconazole [19]. In another example, while voriconazole inhibition of CYP3A4-mediated metabolism of inhaled/intranasal corticosteroids such as fluticasone may result in Cushing syndrome in some patients, alternative corticosteroids less dependent on CYP3A4 metabolism (such as beclomethasone), or switching to isavuconazole, are likely safe [20]. Similarly, while simvastatin and atorvastatin AUCs increase >100% when co-administered with some azoles, empirical dose reductions of 50% with subsequent monitoring is recommended by some authors to be safe. Alternatively, other statins such as pravastatin do not undergo significant CYP3A4-mediated metabolism and thus can be substituted [9]. Finally, while voriconazole-omeprazole combination can result in increased exposure to both agents via competitive CYP2C19 metabolism, administration with pantoprazole is less likely to be problematic [21].

## 3. Adverse Reactions

The extended duration of antifungal prophylaxis predisposes lung transplant patients to a broad range of toxicities. Table 2 provides a summary of azole-associated adverse reactions grouped by organ system.

### 3.1. Hepatic

All azole antifungals can generate some degree of hepatocellular, cholestatic, or mixed liver injury, which can occur at any point during therapy [15,22,23]. The reader is encouraged to reference in-depth reviews of azole-induced hepatotoxicity for additional information [22,23]. Incidence of azole-related hepatotoxicity among lung transplant patients ranges from 5–63% with voriconazole use whereas <10% of patients develop hepatotoxicity on isavuconazole and posaconazole [3,29,30,31,32,33,34,35,36]. When directly compared, voriconazole has demonstrated significantly higher rates of hepatotoxicity versus isavuconazole both for the treatment of invasive mold disease in patients with mostly hematologic diseases (16% vs. 9%, *p* = 0.016), as well as for antifungal prophylaxis in lung transplant recipients (18% vs. 5%, *p* < 0.0001) [33,36]. Moreover, voriconazole-induced elevated liver function enzymes is commonly listed as a reason for azole discontinuation due to side effects. In a study examining reasons for discontinuation of prophylactic azoles in lung transplant recipients, 54.5% of voriconazole exposure episodes resulted in early therapy discontinuation due to side effects, with LFT abnormalities listed as the cause of discontinuation in 18.1% of cases [35]. Fortunately, azole-induced hepatotoxicity is generally considered reversible upon therapy discontinuation or replacement with another azole antifungal. A case report and two retrospective studies demonstrated improvement in liver function test abnormalities after switching from voriconazole to posaconazole in patients requiring continued antifungal therapy [37,38,39]. Thus, transition to either isavuconazole or posaconazole can be considered for patients who develop voriconazole-related hepatotoxicity. A discussion of the correlation, or lack thereof, between azole serum concentrations and hepatotoxicity can be found in Section 4 of this publication.

### 3.2. Central Nervous System and Visual Disturbances

Voriconazole also demonstrates unique central nervous system (CNS) side effects not routinely seen among patients receiving other azoles. The exact mechanism of voriconazole neurotoxicity has not been elucidated. Visual disturbances, comprising blurred or abnormal vision, color vision change, and photophobia, are side effects specific to voriconazole and occur in ~30% of patients [40,41,42,43]. Less common (~4–16%) are voriconazole-induced visual and/or auditory hallucinations [24,40,43,44]. In a study comparing isavuconazole versus voriconazole for antifungal prophylaxis in lung transplant patients, 13 of 151 patients on voriconazole (9%) discontinued therapy due to neurotoxicity versus zero patients on isavuconazole [33]. Both visual disturbances and hallucinations associated with voriconazole use tend to present in the setting of elevated levels (see Section 4 for discussion) and within the first weeks of therapy but resolve after continued use, dose decrease, or drug discontinuation [24,40,41,42,43,44,45,46]. Patients should be warned prior to starting voriconazole of the risks of neurotoxicity as well as reassured that side effects are reversible.

### 3.3. Cardiovascular System

A potential concern post-transplant is the risk of azole-mediated corrected QT interval (QTc) prolongation and subsequent life-threatening cardiac arrhythmias including Torsades de Pointes. In general, drug-induced QTc prolongation is the result of an intrinsic ability to block cardiac hERG-mediated potassium channels resulting in delayed cardiac repolarization and/or mechanisms which increase the exposure of co-administered QTc prolonging drugs [47]. In a study of healthy volunteers, administration of increasing doses of voriconazole showed negligible impact on QT interval with mean changes of less than 10 ms [48]. A similar study conducted among patients on posaconazole found no correlation between increase in QTc and posaconazole exposure [49]. Thus, azoles alone are not considered high-risk for QTc prolongation. Rather, it is the combination of posaconazole and voriconazole with other risk factors, including increased age, electrolyte abnormalities, heart disease, and other QTc prolonging medications, especially those affected by azole-mediated CYP3A4 enzyme inhibition, that may yield an increased risk of serious QTc changes [47,50,51]. Even in such settings, however, most studies are able to elucidate risks for QT prolongation but not for cardiac events such as Torsades de Pointes. For example, in a retrospective study of patients with hematological malignancies, the mean change in QTc from baseline to post-addition of fluoroquinolone-azole combination therapy in 94 patients was 6.1 ms (95% confidence interval (CI) 0.2 to 11.9). One patient experienced a change deemed to be of ‘major clinical significance’, defined as a QTc change of >60 ms or a follow-up QTc measurement of >500 ms, while 20 patients experienced a change of ‘moderate significance’, defined as a QTc change of >30 ms but <60 ms, or a follow-up QTc of >470 ms (men) or 480 ms (women). On univariate analysis, hypokalemia and an ejection fraction <55% were significantly associated with major/moderate clinical changes. No cases of arrhythmia were recorded [50]. In a retrospective study of 46 patients who had received monotherapy with either voriconazole or amiodarone and then subsequent combination therapy, investigators reported a mean change in QTc from baseline of 49.0 ms, with 39.1% of patients having a QTc ≥ 500 ms. In multivariate analysis, lower serum potassium was independently associated with a follow-up QTc ≥500 ms. Again, however, no cardiac events were noted [51]. As such, although azole antifungals have been implicated in case reports of patients developing serious arrhythmias, predicting which patients will progress from an increase in QTc with therapy to an arrhythmia is not feasible, and is both exceedingly rare and perhaps idiosyncratic [52]. In addition, default QT-interval correction formulas in electronic health records may overestimate QTc in patients with prolonged QRS intervals or tachycardia [53]. As such, when initiating posaconazole/voriconazole therapy, while due diligence should be performed to assess for risk factors and to correct modifiable risk factors as feasible (electrolyte abnormalities, concomitant medications that can be changed/discontinued), it is unreasonable to expect patients (especially transplant recipients) be devoid of risk factors when initiating such therapies. In the setting of multiple risk factors, or significant risks such as long QT syndrome, history of drug-induced Torsades de pointes, or concomitant use of dofetilide or sotalol, isavuconazole can be considered as an alternative agent. Isavuconazole simultaneously blocks hERG-mediated potassium and the L-type calcium channels in the myocardium thereby shortening, as opposed to lengthening, the QTc interval [25,54]. In a case report of a lung transplant patient, voriconazole-induced QTc prolongation was reversed after initiation of isavuconazole [25].

### 3.4. Integumentary System

Voriconazole-related photosensitivity and phototoxicity are well described side effects of drug administration [26,55,56,57]. The exact mechanism for voriconazole phototoxicity is not well defined with theories including amassing of a voriconazole metabolite with chromophore properties in skin epidermal layer and voriconazole-inhibition of vitamin A metabolism [57,58,59]. In either case, phototoxoticity acutely manifests as painful erythema on areas of sun- and/or ultraviolet (UV)-light exposed skin, typically occurring within the first year of therapy [26,55,56]. Patients should receive counseling to avoid prolonged sun exposure, wear appropriate sun-protective clothing, and apply broad-spectrum sunscreen to prevent sun damage. Unfortunately, when unchecked, phototoxocity can progress to actinic keratosis and ultimately keratinocyte carcinomas after years of voriconazole use [26]. Solid organ transplant patients in particular are at increased risk for skin cancers compared to the general population, with squamous cell carcinomas (SCC) predominating among lung transplant patients [60,61,62]. In a large cohort study, voriconazole use among non-Hispanic white lung transplant recipients compared to no voriconazole was associated with a greater incidence of SCC with 4–7 months (adjusted hazard ratio (AHR) 1.42, 95% CI 1.16 to 1.73), 8–15 months (AHR 2.04, 95% CI 1.67 to 2.50), and more than 15 months (AHR 3.05, 95% CI 2.37 to 3.91). Male sex, increasing age, second or greater transplant, and history of smoking also yielded additional risk of SCC [62]. Thus, in patients with multiple risk factors for voriconazole-induced phototoxicity, azole alternatives such as isavuconazole or posaconazole may be preferred as neither are currently considered to have a significant effect on the development of phototoxocity and SCC.

### 3.5. Musculoskeletal System

Another toxicity primarily associated with long-term voriconazole administration is the development of periostitis. Periostitis, inflammation of the connective tissue surrounding bone, presents as myalgias, diffuse bone pain, elevations in serum alkaline phosphatase, and exostoses with an onset of months to years after voriconazole initiation [63,64,65]. In vitro, voriconazole has uniquely demonstrated an ability to induce osteoblast activity which may contribute to the development of both periostitis and exostoses [66]. Voriconazole-associated periostitis is also considered to be the result of increased fluoride levels and subsequent accumulation into mineralized tissues including bone [27,67,68,69]. Excess fluoride exposure is likely due in part to the azoles’ chemical structure. Voriconazole contains three, whereas, posaconazole and isavuconazole contains two fluoride atoms [27,69]. When measured in patients on long-term azoles for treatment of coccidioidomycosis, mean voriconazole plasma fluoride concentrations were over double that of posaconazole (9.17 vs. 4.06 µmol/L) and approximately five-fold higher than that of the non-fluoride containing azole itraconazole (9.17 vs. 1.74 μmol/L) [67]. Another study of patients with hematologic malignancies demonstrated a median serum fluoride level of 156.5 μg/L for patients receiving voriconazole, well above the normal range of <30 μg/L and median serum fluoride levels observed in patients on posaconazole (<30 μg/L) [69]. Of those patients who experience periostitis, symptoms are generally reversible upon voriconazole discontinuation, however, complete resolution may take months [27,64,68,69]. Alternatively, voriconazole dose reductions have been effective in reducing symptoms and may be an option in patients who cannot transition to other antifungal therapies [68].

### 3.6. Endocrine System

Lastly, prolonged posaconazole use has been noted to cause toxicity associated with mineralocorticoid excess. As a drug class, azoles exert a pharmacologic effect by inhibiting the enzyme 14-alpha-demethylase and thus preventing the conversion from lanosterol to ergosterol [70]. Posaconazole is hypothesized to additionally inhibit 11-beta-hydoxylase and/or 11-beta-hydroxysteroid dehydrogenase type II both of which result in an accumulation of steroid hormones with mineralocorticoid effects [71,72,73,74,75]. This phenomenon, referred to as posaconazole-induced pseudohyperaldosteronism (PIPH), commonly presents as hypertension, hypokalemia, and alkalosis with low renin and aldosterone levels [28,71,72,73,74,75]. Originally, information on PIPH were limited to case reports [71,72,73,74,75]. However, a recent single-center, retrospective study conducted by Nguyen and colleagues demonstrated PIPH incidence may be as high as 23% [28]. PIPH was found to be associated with older age, hypertension prior to posaconazole initiation, and higher posaconazole levels (see Section 4) [28]. PIPH appears to be reversible either upon dose adjustment or discontinuation [71,72,73,74]. It is unclear at this juncture if PIPH is limited to posaconazole use. Two patient cases of resolution of PIPH after switch from posaconazole to isavuconazole provides clinical evidence to support the safety of this agent, although additional research is needed [74,76].

## 4. Pharmacokinetic Considerations and the Role of Therapeutic Drug Monitoring

Each azole differs in terms of pharmacokinetics and need for therapeutic drug monitoring to optimize drug efficacy and minimize toxicities. Table 3 summarizes therapeutic drug monitoring considerations for isavuconazole, posaconazole, and voriconazole including established serum targets for efficacy and toxicity reported in primary literature.

Significant inter- and intra-patient pharmacokinetic variability is observed with voriconazole use. Figure 1 demonstrates the wide range of reported voriconazole serum trough levels in a sample of lung transplant recipients. Unlike other azoles, voriconazole exhibits non-linear pharmacokinetics, with increasing oral and intravenous doses resulting in non-proportional increases in exposure (as expressed by AUC) [41]. For example, a study of voriconazole in allogenic hematopoietic stem cell transplant recipients showed a 50% increase in dose, 200 mg twice daily to 300 mg twice daily, resulted in a 0.4–7.7-fold increase in serum trough levels [85]. This effect is attributed to saturable hepatic metabolism [41]. Moreover, CYP2C19 has proven to play a significant role in voriconazole exposure. CYP2C19 is highly polymorphic, with varying alleles conferring a spectrum of activity from poor to increased function [86]. Specifically, patients with one or two of the CYP2C19*17 alleles are considered rapid or ultra-rapid CYP2C19 metabolizers whereas the presence of the CYP2C19*2 and/or CYP2C19*3 allele can confer intermediate or poor metabolism [86]. Several studies have identified the presence of the CYP2C19*17 allele with lower serum concentrations compared CYP2C19 normal metabolizers [87,88,89,90]. Conversely, voriconazole serum concentrations are elevated in CYP2C19 intermediate and poor metabolizers compared to normal metabolizers, potentially increasing the risk of drug toxicity [87,88,89,90,91]. Frequency of CYP2C19 phenotypes vary based on racial-ethnic groups, with ultra-rapid metabolizer status seen more frequently in white (31.2%) and African-American populations (33.3%), whereas intermediate and poor metabolizers are more common among the Asian community (43–46% and 14–19%, respectively) [92]. In patients with sub- or supra-therapeutic voriconazole levels, a thorough review of drug–drug interactions, potential absorption issues, and concerns for non-adherence should all be evaluated prior to examination of possible CYP2C19 polymorphisms. If no other factors are identified as the cause of non-therapeutic serum troughs, confirmatory genetic testing may be performed. However, genotyping currently provides limited value in the clinical setting where dose adjustments are based on routine therapeutic drug monitoring.

Highly variable voriconazole pharmacokinetics are of particular concern given the drug’s narrow therapeutic index. One of the first studies to assess the relationship between voriconazole serum levels and efficacy when used for treatment of IFIs was conducted by Pascual and colleagues in 2008. In their analysis, voriconazole trough levels of less than or equal to 1000 ng/mL were associated with greater lack of response to therapy, defined as persistent disease after 14 days of voriconazole treatment, progressive disease after 7 days of therapy or breakthrough IFI, (46% vs. 12%, *p* = 0.02) as well as reduced incidence of complete or partial response (54% vs. 88%, *p* = 0.02) [45]. However, other studies have suggested alternative threshold values or demonstrated no correlation between voriconazole serum trough and efficacy [77,78]. Regarding prophylaxis, in a prospective, observational study of lung transplant recipients receiving voriconazole prophylaxis, a threshold of 1500 ng/mL was proposed as a cutoff for efficacy, as IFIs or colonization were significantly more common in patients with no trough >1500 ng/mL. Importantly, 81% of outcomes were due to new colonization, not invasive infection, and all invasive infections were characterized as anastomotic tracheobronchitis [3]. In 2016, two meta-analyses, which combined treatment and prophylaxis studies, concluded that a therapeutic window could be defined [77,78]. In the analysis by Luong, a threshold of 1000 ng/mL was significantly associated with successful outcome, defined by the respective studies (odds ratio (OR) 1.94, 95% CI 1.04–3.62, *p* = 0.04) [77]. Jin and colleagues defined treatment success as complete response, partial response, stable response or beta-D-glucan value improvement by greater than or equal to 50%. As compared to concentrations >500 ng/mL, they identified concentrations of ≤500 ng/mL as being associated with decreased treatment success (risk ratio 0.46, 95% CI 0.29–0.74, *p* = 0.001) [78]. Given the available data, current Infectious Diseases Society of America guidelines recommend achieving a voriconazole trough of greater than 1000 to 1500 ng/mL for efficacy with trough values less than 1000 ng/mL yielding greater risk for treatment failure [5]. Similar thresholds appear appropriate for voriconazole use in prophylaxis.

Moreover, certain serious voriconazole toxicities, particularly hepatotoxicity and neurotoxicity, may exhibit a concentration-dependent effect. In an early open-label, multicenter study, voriconazole serum concentrations greater than 6000 ng/mL resulted in increased rates of hepatotoxicity (abnormal liver function or liver failure) [93]. However, a retrospective analysis of phase II and III trials failed to find a voriconazole level associated with the incidence of hepatotoxicity [94]. Mitsani and colleagues described a correlation between voriconazole serum troughs and serum aspartate transferase (AST) level yet found no relationship between serum levels and alanine transaminase (ALT), alkaline phosphatase, or total bilirubin [3]. Lastly, in their review of predominantly hematological malignancy patients receiving voriconazole for treatment of invasive fungal disease, Pascual and colleagues reported no significant correlation between a specific voriconazole serum trough concentration and drug-induced hepatotoxicity [45]. It is possible voriconazole serum levels may be elevated in patients with hepatic impairment due to reduced drug metabolism via hepatic CYP enzymes. Nonetheless, current data do not support a threshold for which patients would be at high risk for voriconazole-induced hepatotoxicity. Conversely, a voriconazole serum trough level of less than or equal to 5500 ng/mL has been proposed to minimize the incidence of voriconazole-induced neurotoxicity [45]. Pascual and colleagues reported a statistically significant increase in incidence of encephalopathy with voriconazole serum trough levels above 5500 ng/mL compared to levels less than or equal to 5500 ng/mL (31% vs. 0%, *p* = 0.002) [45]. Interestingly, a specific voriconazole serum trough threshold for CNS toxicity (identified in 3 patients) was not demonstrated among lung transplant recipients on voriconazole prophylaxis [3]. Regarding findings from meta-analyses, Luong and colleagues identified a voriconazole toxicity threshold of 6000 ng/mL (OR 4.60, 95% CI 1.49–14.16, *p* = 0.008) with toxicity events including hepatotoxicity and neurotoxicity as well as gastrointestinal intolerance, cutaneous reactions, cardiotoxicity, and metabolic disturbances [77]. Jin and colleagues identified concentrations less than or equal to 3000 ng/mL as resulting in less hepatotoxicity (risk ratio 0.37, 95% CI 0.16–0.83, *p* = 0.02), and the strongest association for neurotoxicity occurring at a threshold of 5500 ng/mL [78]. Given the erratic pharmacokinetic profile and association between drug levels and efficacy and safety, routine therapeutic drug monitoring is recommended for all patients on voriconazole therapy. An initial level should be obtained approximately five days after therapy initiation. Repeat levels are recommended after dose adjustment, initiation or discontinuation of interacting medications, or other changes in patient condition that may affect drug absorption or metabolism. If dose modification is warranted, adjustments should be mild and incremental due to unpredictable results post-modification.

Posaconazole pharmacokinetics are formulation-dependent. The oral solution demonstrates significant variation in pharmacokinetic parameters based on gastric pH and timing of the dose relative to a meal [79]. In a phase I study posaconazole oral solution AUC was significantly increased with the administration of an acidic carbonated beverage compared to posaconazole alone (AUC 5600 ng h/mL vs. 9610 ng h/mL) [79]. Concomitant administration of a proton pump inhibitor reduced exposures by approximately one-third (AUC 5600 ng h/mL vs. 3700 ng h/mL) and, importantly, the reduction in AUC was not significantly improved with the addition of an acidic beverage (AUC 4180 ng h/mL vs. 3700 ng h/mL) [79]. Furthermore, serum drug concentrations are highest when posaconazole oral solution is administered with a fatty meal, with approximately four-fold reduction in AUC when taken in the fasted state [79]. Lastly, posaconazole oral solution displays saturable absorption, thus divided doses are recommended to increase serum drug concentrations [95]. In comparison, posaconazole delayed-release tablets are not impacted by gastric pH, as the product is formulated to bypass the stomach. It may be taken with a high-fat meal of approximately 70 g of fat to increase AUC by 1.5 fold compared to the fasted state [80]. When directly compared, the posaconazole oral solution consistently displays lower exposures versus the delayed-released tablet, with a switch from oral solution to tablet resulting in as high as an approximate 3-fold increase in posaconazole serum trough concentration [96,97]. Therapeutic levels (as defined below) are achieved in >75% of lung transplant recipients receiving a standard dose of posaconazole tablets (300 mg daily) [96,98], and posaconazole tablets exhibited lower intrapatient variability compared to voriconazole and posaconazole oral solution [96]. As a result, posaconazole tablets are preferred over the oral solution.

Given concerns about poor absorption with posaconazole oral solution and possible sub-therapeutic levels, therapeutic drug monitoring should be performed among patients receiving this product and may also be considered with the delayed-release formulation. Ensuring adequate exposure in patients being treated for invasive infection is advisable. As such, clinicians should target a posaconazole trough goal of greater than 700 ng/mL to ensure efficacy when administered for antifungal prophylaxis. The proposed cut-off was determined by an analysis of two, phase III posaconazole prophylaxis trials among allogenic hematopoietic stem cell transplant recipients and neutropenic patients diagnosed with acute myelogenous leukemia or myelodysplastic syndrome [81,99,100]. Importantly, the target concentration was determined using a composite endpoint of successful clinical outcomes. Clinical failure was defined as the occurrence of proven or probable IFI, a need to administer greater than 5 days of empirical treatment with a systemic antifungal other than the study drug during the primary time period of the study, all-cause mortality, study drug discontinuation during the primary time period, or a patient lost to follow-up. Of the patients with breakthrough IFI, 80% (12/15) had a posaconazole steady-state average serum concentration less than 700 ng/mL [81,99,100]. Smaller studies, including an analysis of lung and heart transplant recipients, have suggested a lower target threshold of 500 ng/mL [101,102]. A serum trough level greater than 1250 ng/mL is recommended for treatment with posaconazole. This threshold was first defined in a trial of posaconazole for treatment of invasive aspergillosis in patients refractory or intolerant to conventional therapy [82]. In the study, average and maximum plasma concentrations were divided into quartiles with the highest quartile average plasma concentration of 1250 ng/mL yielding the greatest response rate, defined as resolution or improvement in clinical signs and symptoms of infection and radiological and mycological abnormalities, of 75% [82]. Historically, posaconazole serum levels were not predictive of toxicity, however, there are emerging data to suggest a correlation between levels and PIPH [28]. Specifically, Nguyen and colleagues demonstrated a positive correlation between incidence of PIPH and posaconazole serum level in patients receiving drugs for treatment or prophylaxis, with low PIPH occurrence (6.5%, 3/46) at serum levels less than 2000 ng/mL, 57% (13/23) PIPH incidence with serum levels greater than or equal to 2000 ng/mL, and 100% (5/5) of patients with serum levels greater than or equal to 4000 ng/mL meeting criteria for PIPH [28]. In summary, while current data suggests a correlation between posaconazole serum levels and incidence of PIPH, additional evidence is needed to recommend a toxicity cut-off at this time. Additionally, while studies have not utilized a single time point for measuring posaconazole concentrations, trough concentrations are recommended, as this would ensure that all concentrations are above thresholds identified. In addition, posaconazole has a half-life of ≥24 h [79], so that while there may not be significant variability at steady-state within an every 6- to 8- hour suspension dosing regimen, variability will be more pronounced within the tablet 24 h interval.

Isavuconazole has demonstrated limited pharmacokinetic variability, thus questioning the necessity of routine therapeutic drug monitoring. Data from the SECURE clinical trial indicates no relationship between exposure and therapeutic response, with greater than 97% of patients attaining serum trough concentrations between 1000 and 7000 ng/mL [83,84]. Similar levels were also observed in a study of solid organ transplant recipients [103]. In their analysis of isavuconazole concentrations in clinical practice, Andes and colleagues found serum levels to be nearly identical to those seen in clinical trials, further supporting the lack of necessity of routine therapeutic drug monitoring with isavuconazole [104]. An analysis of 264 isavuconazole blood concentrations from 19 patients over a median duration of 90 days yielded a suggested target serum trough concentration range between 2500 and 5000 ng/mL [105]. The authors identified no clear threshold for efficacy given low numbers of isavuconazole failure. However, the cut-off for toxicity based on receiver operating curve (ROC) curve analysis demonstrated increases in patient-reported gastrointestinal toxicities at serum levels above 5000 ng/mL [105]. The authors did not identify an association between isavuconazole serum levels and hepatotoxicity assessed via ALT, alkaline phosphatase, and serum total bilirubin [105]. Data on serum targets for prophylaxis is limited. In a retrospective study comparing isavuconazole to voriconazole for antifungal prophylaxis in lung transplant recipients, serum trough levels of the four patients with breakthrough IFIs were 700–1700 ng/mL. However, data on serum levels for patients without breakthrough disease is not available [33]. Given the lack of both pharmacokinetic variability and well-established correlation between drug levels and efficacy or toxicity, routine therapeutic drug monitoring may not be necessary for all patients. It may be considered in unique scenarios such as concomitant administration of CYP3A4 inducers, concerns for toxicity, disease progression, and/or non-adherence. If monitoring is performed, it is reasonable to target trough concentrations of at least 1000 ng/mL to meet typical concentrations seen in clinical trials.

Patients with cystic fibrosis (CF) demonstrate unique pharmacokinetic profiles, including impaired gastric absorption and enhanced drug clearance, that may require special considerations when administering azoles post-lung transplantation. In an analysis of 35 CF lung transplant recipients on voriconazole, 30% of patients had initial levels less than 500 ng/mL [106]. Moreover, in Mitsani and colleagues’ study of voriconazole prophylaxis for lung transplant recipients, CF patients, as compared to non-CF patients, were more likely to have initial voriconazole trough concentrations of less than or equal to 1500 ng/mL based on univariate analysis [3]. Lastly, CF was identified as a risk-factor for sub-therapeutic voriconazole levels using regression analyses in a study including 64 lung transplant recipients, 14 with CF, on voriconazole primarily for treatment [96]. Several studies have also demonstrated reduced posaconazole serum levels in lung transplant patients with CF compared to non-CF patients [96,97,98,107,108]. In a study of 20 patients receiving posaconazole oral solution, serum average posaconazole levels at steady state (median 233 ng/mL vs. 594 ng/mL, *p* = 0.03) and AUC (median 7 vs. 20 h ng/mL, *p* = 0.02) were significantly reduced in the seven patients with CF compared to the non-CF cohort [107]. Furthermore, in a study of posaconazole delayed-release tablets administered to CF lung transplant, non-CF lung transplant, and non-transplant recipients, patients with CF yielded lower mean trough concentrations (1100 vs. 1900 vs. 2400 ng/mL, *p* < 0.00001) compared to other study participants [108]. In another study, when posaconazole oral solution and delayed-release tablet serum levels were assessed in lung transplant recipients, switching from oral solution to tablets resulted in an increase in trough levels, as expected. However, CF was associated with 48% lower trough values in patients receiving the tablet formulation, while no such association was identified for patients receiving oral solution [97]. Similar findings were reported by Stelzer and colleagues [96]. Finally, a case report by Kabulski and colleagues described serum trough levels less than 1000 ng/mL on days 16 and 28 of isavuconazole therapy in a patient with cystic fibrosis after bilateral lung transplant [109]. Such studies emphasize the need for routine therapeutic drug monitoring to guide dose adjustments in this specific patient population.

## 5. Conclusions

As the use of mold-active azole antifungals continues to expand among lung transplant recipients, a thorough understanding of azole pharmacology is necessary to optimize drug safety and efficacy in this patient population. Each azole demonstrates unique drug interactions, short- and long-term adverse drug reactions, and pharmacokinetic properties including need for routine therapeutic drug monitoring. It is imperative these factors, in addition to efficacy data, are taken into account when determining optimal therapy for post-lung transplant antifungal prophylaxis.

## Figures and Tables

**Figure 1 jof-07-00076-f001:**
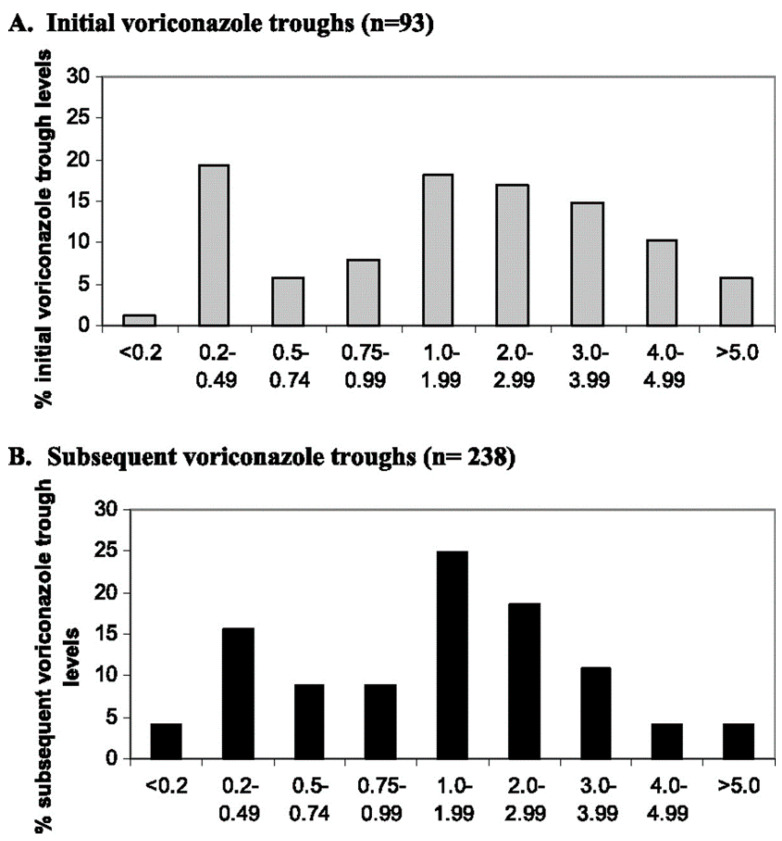
Distribution of voriconazole troughs after initial and subsequent doses in a sample of lung transplant recipients [3]. X-axis displays range of recorded voriconazole trough values in µg/mL. Y-axis describes percentage of patients within the respective voriconazole trough ranges. Reproduced with permission from American Society for Microbiology.

**Table 1 jof-07-00076-t001:** Drug interaction potential and magnitude of effect in lung transplant recipients [9,10,11,12,13].

	CYP2C8/CYP2C9		CYP2C19		CYP3A4		P-Glycoprotein	
	Inhibitor	Substrate	Inhibitor	Substrate	Inhibitor	Substrate	Inhibitor	Substrate
Isavuconazole	−	−	−	−	+	++	+	−
Posaconazole	−	−	−	−	+/++	−	++	++
Voriconazole	++	+	++	++	++	++	−	−

**Table 2 jof-07-00076-t002:** Organ system-based adverse reactions potential and magnitude of effect in lung transplant recipients [22,23,24,25,26,27,28].

	Hepatic	CentralNervous	Cardiovascular	Integumentary	Musculoskeletal	Endocrine
Isavuconazole	+	−	− (*decreases* QT)	−	−	−
Posaconazole	+	−	+/− (may increase QT)	−	−	+
Voriconazole	++	++	+ (increases QT)	++	++	−

**Table 3 jof-07-00076-t003:** Strength of evidence for therapeutic drug monitoring and defined efficacy and toxicity serum targets [3,4,5,6,28,33,41,45,77,78,79,80,81,82,83,84].

	Evidence for Therapeutic Drug Monitoring	Proposed Serum Target for Efficacy		Proposed Serum Target for Toxicity	Guideline Recommendations for Serum Targets		
		Prophylaxis	Treatment		AST	IDSA	ISHLT
Isavuconazole	−/+	*	*	*	2000–3000 ng/mL **	*	*
Posaconazole	Oral suspension: +++ Delayed-release tablet: ++	>700 ng/mL trough	>1250 ng/mL trough	***	Treatment: >1000–1250 ng/mL	*	Prophylaxis: >700 ng/mL Treatment: >1250 ng/mL
Voriconazole	+++	≥1000 ng/mL trough	≥1000 ng/mL trough	≤5500 ng/mL trough	Treatment: 1000–5500 ng/mL ****	Treatment: >1000–1500 to <5000–6000 ng/mL	Prophylaxis and Treatment: 1000–2000 to 4000–5000 ng/mL

Abbreviations: AST, American Society of Transplantation Infectious Diseases Community of Practice; IDSA, Infectious Diseases Society of America; ISHLT, International Society for Heart and Lung Transplantation. * Not defined; ** Levels within this range ensure similar exposure as that seen in clinical trials and have not been shown to correlate with response or toxicity; *** Preliminary data from Nguyen et al. suggests a correlation with posaconazole levels >4000 ng/mL and posaconazole-induced pseudohyperaldosteronism (PIPH); **** Higher target ranges are recommended in setting of poor disease prognosis and/or microbiological evidence of resistance (e.g., elevated minimum inhibitory concentrations (MICs)).

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
