# Peer review of "Review of Pharmacologic Considerations in the Use of Azole Antifungals in Lung Transplant Recipients"

_jof, 2021, doi:10.3390/jof7020076_

Round 1

Reviewer 1 Report

This review is very interesting and informative, well written. Therefore, I will comment on what I noticed.

Minor comments

In Table 3.

Blood concentration target values for PSCZ and VRCZ are misleading. You should describe the range with reference to some guidelines and meta-analysis.

This review is a paper on lung transplantation. Therefore, it is better to mention azole-resistant Aspergillus or others.

Author Response

JOF-1064393: Responses to Reviewers

To the editor and reviewers,

Thank you for your detailed review and recommendations. We believe that this input has greatly improved our manuscript. Below are our point-by-point responses to reviewer suggestions. Please note, also, that we made other minor revisions to the text (these can be observed with track changes version of the document).

Kind regards,

Megan Klatt and Gregory Eschenauer.

Reviewer #1:

  1. In Table 3. Blood concentration target values for PSCZ and VRCZ are misleading. You should describe the range with reference to some guidelines and meta-analysis

We agree with the reviewer, and have edited Table 3 for clarity as well as added a column to provide the International Society for Heart and Lung Transplantation, American Society of Transplantation Infectious Diseases Community Practice, and Infectious Disease Society of America guideline recommendations.

  1. This review is a paper on lung transplantation. Therefore, it is better to mention azole-resistant Aspergillus or others.

Although the paper by Shalhoub Journal of Antimicrob Chemother 2015 (which also cites TRANSNET data) suggests a low rate of azole-resistant Aspergillus in lung transplant recipients, we have added a sentence in the Introduction to address this concern. Thank you for the suggestion!

Reviewer 2 Report

To make the review also more accessible for the not clinical focused researcher, I would like to suggest extending the introduction. The following state-of-the-art aspects could be included:

  • Which Aspergillus species are responsible for infections in lung transplant recipients?
  • Besides Aspergillus, what are other pathogenic yeast species frequently observed?
  • What are the differences/similarities between yeast infections in lung transplant recipients compared to other types of yeast infections?
  • Discuss the azole antifungals (voriconazole, posaconazole and isavuconazole) that were selected for this review, in relation to the other azole antifungals, and other types of antifungals (compare also their mechanisms of action). What are the reasons why these azole antifungals are selected to be discussed in this review?

Minor remarks

  1. Page 2, line 45: Table caption should be in sentence case; the same remark applies for the other captions.
  2. Page 2, Table 1: Its is not clear why for the column “P-glycoprotein” the effect is indicated with “yes”, “yes (weak)” and “no”, and in the other columns the effect is indicated with “+”, “++” and “-“
  3. Page 2, line 75: Explain the abbreviation “AUC” the first time it is encountered in the text.
  4. Page 3, line 107: Suggested to remove “previously published in the literature” from the sentence since it is redundant information.
  5. Page 3, line 126: … can be found in the therapeutic drug monitoring section … => give the title of this section
  6. Page 4, line 150: remove “milliseconds” and keep “ms”.
  7. Page 5, line 205: explain the abbreviations “AHR” and CI”
  8. Page 5, line 227: … induce osteoblast activity … => not clear: Are osteoblasts stimulated and osteoclast activity inhibited?
  9. Page 6, Table 3 and further in the text: use “µg/ml” instead of “mcg/ml”.
  10. Page 6, Table 3: info under the table: Walsh et al. refers to which reference number? “***Preliminary data …”: Does this refer to own not yet published research?

11. Page 8, Figure 1: Indicate the figure caption below figure. Include a more detailed explanation of the figure in the figure caption. What are the values (and units) in the x axis representing?

Author Response

JOF-1064393: Responses to Reviewers

To the editor and reviewers,

Thank you for your detailed review and recommendations. We believe that this input has greatly improved our manuscript. Below are our point-by-point responses to reviewer suggestions. Please note, also, that we made other minor revisions to the text (these can be observed with track changes version of the document).

Kind regards,

Megan Klatt and Gregory Eschenauer.

Reviewer #2:

To make the review also more accessible for the not clinical focused researcher, I would like to suggest extending the introduction. The following state-of-the-art aspects could be included:

  • Which Aspergillus species are responsible for infections in lung transplant recipients?
  • Besides Aspergillus, what are other pathogenic yeast species frequently observed?
  • What are the differences/similarities between yeast infections in lung transplant recipients compared to other types of yeast infections?
  • Discuss the azole antifungals (voriconazole, posaconazole and isavuconazole) that were selected for this review, in relation to the other azole antifungals, and other types of antifungals (compare also their mechanisms of action). What are the reasons why these azole antifungals are selected to be discussed in this review?

We agree with the reviewer in providing additional details on IFIs in lung transplant recipients in the Introduction. The section describing fungal pathogens was expanded to include Aspergillus fumigatus as the predominant species of Aspergillus responsible for infections in this population. Information was also added to describe the possible risk of azole-resistant Aspergillus spp. and the occurrence (and type) of IFIs due to Candida spp., particularly Candida albicans.

In regards to the discussion on the agents selected for this review, the second paragraph describes this rationale. We have added a rationale for the lack of inclusion of the echinocandins.

Page 2, line 45: Table caption should be in sentence case; the same remark applies for the other captions.

Our current formatting is consistent with that of past reviews in Journal of Fungi.

Page 2, Table 1: Its is not clear why for the column “P-glycoprotein” the effect is indicated with “yes”, “yes (weak)” and “no”, and in the other columns the effect is indicated with “+”, “++” and “-“

Adjusted Table 1 to include “+”, “++”, and “-“ instead of “yes” “yes (weak)” and “no” for the P-glycoprotein column.

Page 2, line 75: Explain the abbreviation “AUC” the first time it is encountered in the text.

Thank you, we have added an explanation for the abbreviation.

Page 3, line 107: Suggested to remove “previously published in the literature” from the sentence since it is redundant information.

Thank you, we have made this change.

Page 3, line 126: … can be found in the therapeutic drug monitoring section … => give the title of this section

Thank you, we have made this change.

Page 4, line 150: remove “milliseconds” and keep “ms”.

Thank you, we have made this change.

Page 5, line 205: explain the abbreviations “AHR” and CI”

Thank you, we have made these changes.

Page 5, line 227: … induce osteoblast activity … => not clear: Are osteoblasts stimulated and osteoclast activity inhibited?

This in vitro study did not examine the effect of voriconazole on osteoclasts. However, we did move this sentence to earlier in the section, where other mechanisms are discussed.

Page 6, Table 3 and further in the text: use “µg/ml” instead of “mcg/ml”.

Thank you, we have made this change.

Page 6, Table 3: info under the table: Walsh et al. refers to which reference number? “***Preliminary data …”: Does this refer to own not yet published research?

We have removed the information from Walsh et al, as we do not feel it to be absolutely necessary and is potentially confusing. “Preliminary data” refers to the information presented by Nguyen and colleagues which has yet to be replicated in a robust manner. Added direct reference to this paper in the comments for clarification.

Page 8, Figure 1: Indicate the figure caption below figure. Include a more detailed explanation of the figure in the figure caption. What are the values (and units) in the x axis representing?

Moved caption to below figure and included a more detailed explanation of the Figure.
